# The Preparation of Stewards with the Mastery Rubric for Stewardship: Re-Envisioning the Formation of Scholars and Practitioners

**Christopher M. Rios [1], Chris M. Golde [2] and Rochelle E. Tractenberg [3],*** 

[1] Graduate School, Baylor University, Waco, TX 76798, USA; chris_rios@baylor.edu
[2] BEAM Stanford Career Education, Stanford University, Palo Alto, CA 94305, USA; golde@stanford.edu
[3] Collaborative for Research on Outcomes and –Metrics, Departments of Neurology, Biostatistics, Bioinformatics & Biomathematics, and Rehabilitation Medicine, Georgetown University, Washington, DC 20057, USA
* Correspondence: rochelle.tractenberg@gmail.com

**Abstract:** A steward of the discipline was originally defined as "someone who will creatively generate new knowledge, critically conserve valuable and useful ideas, and responsibly transform those understandings through writing, teaching, and application". This construct was articulated to support and strengthen doctoral education. The purpose of this paper is to expand the construct of stewardship so that it can be applied to both scholars *and non-academic practitioners*, and can be initiated earlier than doctoral education. To accomplish and justify this, we describe a general developmental trajectory supporting cross-curriculum teaching for stewardship of a discipline as well as of a *profession*. We argue that the most important features of stewardship, comprising the public trust for the future of their discipline or profession, are obtainable by all practitioners, and are not limited to those who have completed doctoral training. The developmental trajectory is defined using the Mastery Rubric construct, which requires articulating the knowledge, skills, and abilities (KSAs) to be targeted with a curriculum; recognizable stages of performance of these KSAs; and performance level descriptors of each KSA at each stage. Concrete KSAs of stewardship that can be taught and practiced throughout the career (professional or scholarly) were derived directly from the original definition. We used the European guild structure's stages of Novice, Apprentice, Journeyman, and Master for the trajectory, and through a consensus-based standard setting exercise, created performance level descriptors featuring development of Bloom's taxonometric cognitive abilities (see Appendix A) for each KSA. Together, these create the Mastery Rubric for Stewardship (MR-S). The MR-S articulates how stewardly behavior can be cultivated and documented for individuals in any disciplinary curriculum, whether research-intensive (preparing "scholars") or professional (preparing members of a profession or more generally for the work force). We qualitatively assess the validity of the MR-S by examining its applicability to, and concordance with professional practice standards in three diverse disciplinary examples: (1) History; (2) Statistics and Data Science; and (3) Neurosciences. These domains differ dramatically in terms of content and methodologies, but students in each discipline could either continue on to doctoral training and scholarship, *or* utilize doctoral or pre-doctoral training in other professions. The MR-S is highly aligned with the practice standards of all three of these domains, suggesting that stewardship can be meaningfully cultivated and utilized by those working in or outside of academia, supporting the initiation of stewardship prior to doctoral training and for *all* students, not only those who will earn PhDs or be scholars first and foremost. The MR-S can be used for curriculum development or revision in order to purposefully promote stewardship at all levels of higher education and beyond. The MR-S renders features of professional stewardship accessible to all practitioners, enabling formal and informal, as well as self-directed, development and refinement of a professional identity.

**Keywords:** Mastery Rubric; stewardship; curriculum development and evaluation; actionable evidence of learning; professional identity; professional development

---

## 1. Introduction

In 2001, the Carnegie Foundation for the Advancement of Teaching instituted a five-year project entitled the "Carnegie Initiative on the Doctorate" (CID) that sought to examine the purpose of doctoral education in the United States. The primary outcomes included two publications: *Envisioning the Future of Doctoral Education: Preparing Stewards of the Discipline* [1] and *The Formation of Scholars: Rethinking Doctoral Education for the Twenty-First Century* [2]. These volumes argued that the goal of doctoral education was to create more than simply experts in a field but also stewards of a discipline [1] (p. 5). The CID defined a steward as "a scholar first and foremost", one to whom "we can entrust the vigor, quality, and integrity of the field" (p. 5) and went on to describe the activities of a steward in three categories—generation, conservation, and transformation. A steward is someone who "will creatively generate new knowledge, critically conserve valuable and useful ideas, and responsibly transform those understandings through writing, teaching, and application" [1] (p. 5).

In a higher-education symposium held at Baylor University in October 2016, a panel reviewed the first 10 years of the construct's use [3] and drew attention to current applications in three settings: in the Graduate School at Baylor University (Waco, Texas), the "Reframing the PhD" project (2015–2018) in Australia (http://reframingphd.com.au/), and as an alternative model for training and certifying "mentors" through the American Statistical Association. The panel affirmed the validity and value of the stewardship model as the goal of doctoral education, demonstrating how the active promotion of disciplinary stewardship is well worth supporting [3–5]. In doing so, they also highlighted the relevance of the characteristics of stewardship in areas beyond the scope intended by the CID. Namely, they asserted: (1) its validity for professionals and practitioners within a discipline as well as scholars in the traditional sense; and (2) its validity for educational levels prior to the doctorate. This expansion is alluded to in the original [1] (p. 10): "Upon entry into practice, all professionals assume at least a tacit responsibility for the quality and integrity of their own work and that of colleagues. They also take on a responsibility to the larger public for the standards of practice associated with the profession". In order for "all professionals" to assume these responsibilities, teaching and learning about them is essential. The purpose of this paper is to expand the construct of stewardship so that it can be developed by all professionals and practitioners, and can be initiated earlier than doctoral education.

This paper expands upon the ideas generated at the Baylor symposium. We developed a general curricular framework that can be used by different disciplines to promote the explicit integration of stewardship throughout higher education and into the workplace. This framework is in the form of a curriculum development tool that can also be used outside of formal curricula, the Mastery Rubric [6]. With its development, we sought to ensure that, rather than assuming a "tacit responsibility" for integrity and their professional practice standards, all those who proceed through the developmental trajectory for stewardship—whether or not they do so in a doctoral curriculum—would be prepared to engage fully and responsibly in their profession.

Articulating such a framework adds formative value to the stewardship model by permitting both students and faculty to determine where they are in their own stewardly development so that individuals can set goals for further growth. Additionally, while we agree that the characteristics of a steward are fully and most explicitly formed through doctoral education, many individuals do, or would, advance and sustain their disciplines *without* pursuing doctoral education. Further, many doctorally-prepared faculty around the US *teach* first and foremost, with scholarship playing a secondary role in their professional lives. Thus, the construct *is* relevant for more than just "the scholar first and foremost". We therefore argue for the expansion of the stewardship model beyond doctoral education and beyond those for whom scholarship is their primary responsibility. This expansion

could also enhance the practical value of the model, but articulating a developmental trajectory for stewardship is needed to promote its wider adoption across disciplines and levels of practice, providing guidance for curricula that can prepare stewards throughout higher education rather than solely for doctoral education.

Subsequent sections of the Introduction more fully describe stewardship and explain and contextualize the proposed expansion of its scope. The Mastery Rubric construct is also introduced. The Methods Section is focused on the creation of the Mastery Rubric for Stewardship and the validity evidence supporting its creation and use, as well as the alignment of this Mastery Rubric with diverse disciplinary professional practice standards. The results of these methods reinforce the determination that stewardship can be achieved by all practitioners, and can be initiated prior to doctoral training.

### 1.1. Who is A Steward of the Discipline?

As noted, the CID described stewards of the discipline as those committed to the foundation and future development of one's field as expressed in three activities: generation, conservation, and transformation. Generation is the heart of doctoral study, and of the PhD in particular. Of all graduate education, the PhD is distinguished by the expectation and obligation to make a new and significant contribution to one's field. Doctoral graduates are expected to be able "to ask interesting and important questions, to formulate appropriate strategies for investigating these questions, to conduct investigations with a high degree of competence, to analyze and evaluate the results of the investigations, and to communicate the results to others to advance the field" [1] (p. 10). Included here is the responsibility to critically evaluate new and existing claims in order to ensure the quality of others' work in the field, and to help promote scholarship that *advances the field*, and does not simply augment the author's CV. Thus, "generation" implies both making one's own contribution and judging the contributions of others.

Conservation means understanding the history and fundamental ideas of one's field so that depth of knowledge in one aspect of the discipline is balanced by understanding how that knowledge fits within the discipline overall. While doctoral students typically acquire narrow expertise, their specialization should be balanced by a commitment to knowing the field more broadly. Similarly, stewards know how their field (or their niche in it) complements the larger intellectual landscape beyond their own discipline. Stewards are thereby called to move the field forward while maintaining its defining characteristics. By doing so, stewards are "aware of the shoulders on which they stand and are able to judge which ideas are worth keeping and which have outlived their usefulness" [2] (p.12).

Transformation includes the tasks of responsible writing, teaching, and application. It represents "teaching in the broadest sense of the word" [2] (p.12). Whether one is in or outside of a traditional academic or research setting, stewards must clearly communicate within and across traditional disciplinary boundaries and to diverse audiences, including novice learners, specialists in the field, or the broader society. These transformations represent significant achievements, not all of which are taught or practiced in doctoral training. Transformation also includes the ability to apply one's knowledge and expertise to help solve problems, or bring greater understanding, within and outside of the specific discipline. Such application varies by field and topic of study, but stewards in all fields have a responsibility for transparency—open, honest, and thorough documentation or communication in all aspects of this broadest sense of teaching, including writing, teaching, mentoring, and the application of the knowledge of their discipline to problems within and outside of their discipline.

Thus, stewards are those who contribute to the generation, conservation, and transformation of their fields, and PhD program graduates should be able to bring their knowledge, skills, and expertise to bear on a wide range of challenges [7]. Significantly, the CID did not limit stewards to traditional faculty roles, but affirmed the diverse career paths pursued by doctoral graduates (see also, e.g., [8–10] (p. 299). Whether they serve in the academy as researchers or teachers, or they use their education in business, government, or non-profits, doctoral graduates may be seen as scholars, because, in the words of the CID, "the work of scholarship is not a function of setting but of purpose and commitment" [2] (p. 8). Admittedly, this is a broad view of scholarship, stemming in part from the Carnegie Foundation's

understanding of doctoral education as a kind of professional preparation: it is preparation for an academic profession [2] (pp. X–XI). This broad view is one we affirm, in so far as "scholar" means a steward of the discipline. Where we differ from the CID is with their: (a) view that stewardly formation occurs only within doctoral education; and (b) definition of stewards as those for whom scholarship is first and foremost. That is, we argue that, while stewards are fully and explicitly formed at the doctoral level, stewardship *can* also be developed earlier than doctoral training, and/or in professional practice. In our model, instilling the characteristics and identity of a steward can be a goal for doctoral, professional, and undergraduate education, promoting a commitment to the integrity and vigor of the discipline by scholars, professionals, and practitioners within a field.

For example, in many professions, conservation is practiced in that there are requirements for an understanding of the fundamental ideas of the field. Nurses, accountants, lawyers, statisticians, and many other professionals have a responsibility to maintain the standards and fundamental principles of their professions. Similarly, transformation in the words of the CID, "teaching in its broadest sense", is a common expectation across workplaces, professions, and environments. Even "generation"—the hallmark of the PhD—can be achieved by those who are dedicated to their fields yet lack the doctoral degree. Concretely illustrating this point is the GitHub repository for code and computing tools, which is a venue for "publishing" and making public—open to user and community input—new tools and techniques to promote scientific advances (see, e.g., [11]). Contributors to resources like GitHub need not hold doctorates, but they *do* need to be stewardly (e.g., [12]).

Additionally, the rate of PhD production across disciplines in the US has long outstripped the needs of the academic job market (see, e.g., [13]). Thus, increasing numbers of doctoral graduates work outside of the academy, and yet may continue to function as stewards in their non-scholarly roles. In addition, many doctoral graduates seek employment in colleges or universities where *teaching*, and not scholarship, is the principal obligation. If the construct of stewardship is limited to those for whom scholarship is in fact "first and foremost", then the majority of college and university faculty (whose principal role is education, not scholarship) might not self-identify as stewards—which is clearly neither desirable nor true. Importantly, modern scientific practice has become highly inter- and multi-disciplinary, which suggests that academics may be contributing scholarship to their own discipline as well as others, or into the literature that comprises the intersection(s) between disciplines. These recent changes in the landscape of scholarly preparation and employment must be accommodated if the construct of stewardship is to be more widely engaged. We seek to promote commitment to the construct in the widest possible sense in scholars, educators, and professionals outside of the academy.

Finally, reserving the cultivation of stewardship to only doctoral students, who are the smallest proportion of students in any academic setting (sometimes by orders of magnitude), *leaves to chance* the development of stewardly attitudes towards a discipline or profession among the vast majority of graduates. While it is argued that "(u)pon entry into practice, *all professionals assume at least a tacit responsibility* for the quality and integrity of their own work and that of colleagues" [1] (p. 10—emphasis added), this should not describe only professionals who are graduates of doctoral programs. We conclude that, for the benefit of society broadly, and higher education specifically, the stewardship model *should not be limited* to doctoral education.

## 1.2. A Developmental Path for Stewardship

In the words of the CID, "doctoral education is a complex process of formation". It includes technical training, but is more importantly concerned with developing the intellectual expertise, commitments, and perspectives required of a custodian of the field. "What is formed, in short, is the scholar's professional identity in all its dimensions" [2] (p. 8). This is perhaps most true—and observable—in doctoral education, but professional identity development could be more widely promoted if stewardly preparation were started *earlier* than the doctoral level. A Mastery Rubric is a tool for curriculum development and evaluation [6] that articulates a curriculum's intended outcomes and integrates a developmental trajectory that moves learners from novice to more expert performance

within an evidence-centered design framework. The Mastery Rubric construct supports the articulation of a developmental framework for stewardship where students as early as the first year of college (two- or four-year programs) and faculty can engage in, and actively monitor, development of the target knowledge, skills, and abilities [14]. Thus, we used the Mastery Rubric construct to create a developmental path for stewards of the discipline, profession, or enterprise across disciplines.

Mastery Rubrics have been published for clinical research [14], ethical reasoning [15], evidence-based medicine [16], and statistical literacy [17]. The construct is described in detail in [6]. In order to create a Mastery Rubric (MR), three elements are needed [6]:

(1). A list of **k**nowledge, **s**kills and **a**bilities (**KSAs**) to be developed by learners via the curriculum in order for progress towards expertise (or independence) to be attained;
(2). Conceptualization of the **trajectory** of progression from novice to the desired (targeted) level(s) of expertise or independence, and ideas of the evidence supporting claims of classification; and
(3). Descriptions of two or more **mutually exclusive performance levels** (e.g., novice, proficient) on *each* of the list of KSAs.

Performance of the KSAs that are the ultimate objective of the curriculum should progress in recognizable stages from uninitiated towards expertise or independence. The specific achievements of a person at each stage and for each KSA are represented in the Mastery Rubric as performance level descriptors (PLDs). PLDs clarify, but do not restrict or prescribe, what instructors need to teach and assess, and what students need to demonstrate in order for their performance of a KSA at a given stage to be recognized as "achieved". The MR should capture the consensus around the curriculum; making the KSAs explicit, and their achievement at target stages explicit, so decisions about a new curriculum or revisions to existing ones can be facilitated and communicated.

## 2. Methods

To develop a Mastery Rubric for Stewardship (MR-S, Table 1), we completed each of the three elements, described below. We then sought to determine the value and utility of the construct through three case studies, assessing the alignment of the MR-S KSAs with professional practice standards, all of which are relevant for practitioners of that field in or outside of academia, by way of a Degrees of Freedom Analysis [18,19].

**Table 1.** The Mastery Rubric for Stewardship.

| Developmental stage/ performance level of *stewardship* Stewardship KSAs | Novice | Apprentice | Journeyman | Master |
|---|---|---|---|---|
| *General descriptor of performance:* | Has interest but limited experience in the discipline or profession, but is being introduced to the ideas and commitments that the Apprentice will build upon.<br><br>Is discovering the importance of disciplinary and professional stewardship. | Actively engaged in study of the discipline and seeks opportunities to demonstrate and grow the KSAs.<br><br>Developing the full range of Bloom's cognitive abilities, a greater awareness of their own limitations, and a commitment to professional and disciplinary stewardship. | Demonstrates the KSAs and commitments of a steward of the discipline, including preserving disciplinary integrity.<br><br>Is engaged in a disciplinary or professional community, and seeks additional opportunities to reinforce less-well developed skills. | An expert in the KSAs and someone to whom apprenticeship in stewardship can be entrusted.<br><br>Formatively diagnoses and remediates the performance of KSAs, and develops and evaluates summative assessments for specific KSAs in support of stewardly development through the master level. |
| Requisite knowledge/ situational awareness | Largely unaware of the professional community and standards within which their academic or professional interests operate. | Learning to recognize when and how to demonstrate stewardship, that professional standards of practice involve both legal/illegal and ethical/unethical continua, and how to recognize and respond to these features. | Exercises professional practice standards and recognizes situations in which stewardship should be modeled and/or applied with respect to themselves and others, and to interactions within and outside of the profession or discipline. | Models, promotes, and teaches recognition of situations in which stewardship can and should be demonstrated; identifies strategies for how best to proceed when it is not clear. |
| Create and/or generate new methods/new knowledge | Has limited awareness of the knowledge and activities of the discipline, and limited exposure to the ethical issues involved in their creation and use.<br><br>Learning that knowledge is *generated*; and that the creation of new methods or knowledge may have ramifications beyond the original intent. | Learning to create methods and knowledge in a manner that strengthens and advances the field and disciplinary community.<br><br>Developing the ability to recognize when new methods or knowledge can be used for unethical ends, and how stewards of the discipline respond.<br><br>Learning how to balance a commitment to strengthen and advance the discipline with advancing one's career | Generates, and transparently communicates, new methods and knowledge to strengthen and advance the field.<br><br>Considers how new ideas can be used for unethical ends, and models how to respond when such action occurs.<br><br>Prioritizes the disciplinary community over metrics that devalue it. Challenges such metrics whenever possible. | Models, promotes, and teaches stewards to recognize and exhibit their responsibilities to the disciplinary community and society as they create and/or generate new methods and knowledge.<br><br>Promotes transparency in the documentation of the new knowledge/methods to others in the disciplinary community and those outside it.<br><br>Supports systems for professional assessment and developmental milestones for themselves, their mentees/trainees, and others in the community that are consistent with stewardly responsibilities. |
| Critically evaluate extant and emerging ideas | Limited ability to evaluate ideas or differentiate between assertions and arguments within the discipline.<br><br>Uncritically treats vetted ideas and materials as "true".<br><br>Learning how warrants and claims function together to form arguments and evidence-based reasoning. | Learning how professionals review, critique, and challenge each other's ideas and arguments.<br><br>Practicing these skills through guided work with increasing levels of disciplinary engagement and independence.<br><br>Developing the intellectual humility necessary to critically evaluate their work according to disciplinary standards. | Critically evaluates knowledge and ideas within the discipline or profession and the paradigms by which this knowledge is derived, and promotes this evaluation by others.<br><br>Participates in the vetting of new and emerging ideas within the profession or discipline.<br><br>Exhibits intellectual humility and ensures their contributions to the field are well-reasoned and well-supported. | Models, promotes, and teaches critical thinking. Trains stewards in intellectual humility and the critical evaluation (vetting) of extant and emerging ideas, including their own work and the work of others both in and outside of their own discipline. |

**Table 1.** *Cont.*

| Developmental stage/ performance level of *stewardship* Stewardship KSAs | Novice | Apprentice | Journeyman | Master |
|---|---|---|---|---|
| Conserve ideas (or rejects ideas if non-conservation is justified) | Entering the field by learning about the fundamental ideas, thinkers, and accomplishments of the past. Attention is focused on remembering and understanding core (highly conserved) ideas; justifies neither their conservation nor rejection of ideas or arguments. | Learning to conserve fundamental ideas of the field through engagement, application, relation, and extension, as well as qualification and critique. Learning to recognize processes by which ideas in the field are vetted and that re-evaluation and conservation are essential to the integrity of the field. Learning to describe and justify decisions about conservation or non-conservation. Learning how these decisions have shaped the history of the field and the ideas that are/have been conserved. | Critically conserves the ideas that advance the field and preserve its integrity. Recognizes multiple perspectives, including cultural and extra-disciplinary influences, in describing and justifying decisions of conservation or non-conservation of ideas, models, and views. Recognizes their role in shaping the field/ profession and its history. | Trains stewards to recognize, understand, and critically evaluate the vetting that ideas in the field have/have had, including the influence of cultural and extra-disciplinary forces. Models, promotes, and teaches that conservation or non-conservation of ideas in the discipline or profession must be justified, and how to do so. Strives to instill in others, both in and outside of their own discipline, an understanding of the dynamics of the evolution of the field. Participates in the conservation, non-conservation, or rejection of ideas through teaching or training and enabling others to do so. |
| Responsibly write | Learning disciplinary writing standards with attention to the details of what must be recorded, how to construct written reports, and why responsible writing requires transparency. | Gaining greater proficiency in discipline-specific writing. Demonstrating increased sophistication in writing, including content, rhetoric and argumentation, and transparency and professional integrity. | Independently writes in the diversity of contexts and styles specific to the field, to generate, conserve, challenge, and reject field-specific knowledge and to engage others in and outside the field. Practices and promotes transparency in their writing for the sake of the discipline and field. | Trains stewards in the importance and execution of transparent, complete, and appropriate—responsible—writing within the field. May also train stewards in cross-disciplinary writing, and in writing for readers outside the discipline. |
| Responsibly teach/mentor/model | As someone uninitiated in the field, the Novice does not undertake teaching or mentoring roles. | Growing in disciplinary knowledge, skills, and abilities and the ability to pass these on to others. Develops an understanding of the importance of competent mentoring and modeling inherent in professional practice and disciplinary responsibilities. Seeks opportunities to learn about workplace appropriate teaching and learning, and to practice teaching, with supervision if available. | Possess the knowledge, skills, and abilities of the discipline and is able to pass these along to others. Teaches, mentors, and models professionalism and the commitments of stewardship in both formal and informal settings, within and outside the field. | The Master steward teaches others to teach, model, and mentor professionalism and stewardship. * |

**Table 1.** *Cont.*

| Developmental stage/ performance level of *stewardship* Stewardship KSAs | Novice | Apprentice | Journeyman | Master |
|---|---|---|---|---|
| Responsibly apply disciplinary knowledge | As someone inexperienced in the KSAs of the field, the Novice is not expected to apply them but is learning that application entails responsibilities for the practitioner. | Learning how and when to apply the KSAs of the discipline and how application entails professional and ethical responsibilities, including integrity, transparency, and respect. Seeks opportunities to deepen their knowledge of their professional and ethical responsibilities. | Applies the KSAs in a way that preserves and advances the field by demonstrating integrity, transparency, and respect in interactions within and outside of the profession or discipline. | Trains stewards to responsibly apply disciplinary knowledge within and outside of the professional or discipline. Teaches, models, and promotes the recognition and acceptance of the responsibility that accrues to those who practice in the discipline or profession. |
| Responsibly communicate | Discovering the rules for communicating in the discipline or profession. Learning that stewards of the discipline have a responsibility to represent their field to others in a way that promotes the integrity, transparency, and respect of their profession. | Learning how and when to communicate with insiders and outsiders about their discipline or profession. Recognizes that communicating as a steward imparts responsibilities that include demonstrating integrity, transparency, and respect, and seeks to exhibit their commitment to these responsibilities. | Clearly and effectively communicates the ideas, perspectives, and content of the discipline to insiders and outsiders in a way that promotes integrity, transparency, and respect. | Trains stewards to communicate responsibly across modalities and audiences. Teaches, models, and promotes the acceptance of the responsibility that accrues to those who communicate about the discipline or profession, or its results. |

Note: * This cell encompasses the entire MR-S, as well as the entire definition of the steward. Many independent practitioners (journeymen) have responsibilities to mentor or instruct, but have not accumulated evidence of qualification and the focus on the diagnosis and remediation of challenges that are encountered earlier in development, which distinguish the Master.

### 2.1. KSA Identification

The first step in any MR is to identify core KSAs. We started by applying a cognitive task analysis (CTA; [20]) to the Carnegie Foundation's original definition of stewardship. This analysis is outlined in the Supplementary Materials. KSAs were taken directly from the main characteristics of stewardship: generation, evaluation, conservation, and responsibility in disciplinary/scholarly writing, teaching, application, and communication. Since the MR is a tool for curriculum development, the "teachability" of KSAs, and observability of learning, are prioritized; thus, KSAs were revised until these features were realized. KSA articulation was also informed by whether performance at a given stage in a developmental trajectory could be demonstrated concretely, within a variety of curricula, for the specific KSA. The first draft of KSAs were further separated where this brought clarity to the performance level descriptions, and/or where the evidence supporting achievement of different KSAs could plausibly be separable (by job description or by intention). Ongoing discussion led to consensus on the final KSAs. This process is shown in the Supplementary Materials.

### 2.2. Trajectory Articulation

Step 2 in developing a Mastery Rubric is the articulation of the stages along the trajectory. The trajectory is designed to ensure that each KSA is learnable and improvable, with concrete opportunities for assessment and demonstration, at each stage. As with all but one MR (see [6]), we used the European guild structure (see [21] (p. 182)), which identifies novice, apprentice, journeyman, and master stages or levels. These developmental stages conveniently map onto higher education generally, as well as to many professions. Thus, this trajectory can be used for curriculum development or evaluation so that the MR-S can be implemented across educational contexts, but can also be implemented in the workplace.

### 2.3. Performance Level Descriptors

The third step in developing a MR is to describe what each KSA "looks like" when performed at a given stage. Because stewardship KSAs cannot be "tested" but must be observable, we sought to formally specify the level of KSA performance that would be minimally required for an individual to be classified into a given stage on that KSA (following [22], p. 4). The first and last authors did this using a formal approach to performance level description (PLD), combined with the assumption that the performance of a KSA by someone at a given stage can be described using the appropriate level of Bloom's taxonomy [23] (see Appendix A) required for the demonstration or performance of the KSA. Bloom's taxonomy is one of the most widely used, empirically and theoretically supported taxonomies for cognitive functions and is featured in every MR to date. Rather than rely on age, career stage or other such criteria, we drafted *Range* PLDs [24] (pp. 91–92) that could describe the complex behaviors each KSA represents. The three co-authors, who come from different disciplinary perspectives, were participants in the Stewardship panel in 2016 specifically because of their expertise in the development (CMG) and application (CMR and RET) of the construct of stewardship; the first and last authors served as the "subject matter experts" in this iterative standard setting (PLD drafting) exercise following a combination of approaches articulated by Kingston and Tiemann (2012) [25]. Since our goal was to develop a tool that could be used by diverse disciplines, we required that PLDs entailed teachable behaviors that would be demonstrable within a variety of curricula. PLDs were articulated through an iterative process using a modified Body of Work procedure [25]: a first draft of each PLD for a given KSA was created based on Bloom's taxonomy by one co-author (RET), and served as the basis for "range finding" for performance of the KSA at a given stage. PLD drafting used Bloom's taxonomy, refined by appealing to the elements of assessment validity outlined by Messick (1994) [26]:

(1). What is/are the knowledge, skills, and abilities (KSAs) that students should possess at the end (*or a given stage*) of the curriculum?

(2). What actions/behaviors by the students will reveal these KSAs?

(3).　　What tasks will elicit these specific actions or behaviors?

The integration of Bloom's taxonomy and the Messick criteria facilitated the inclusion of concrete and observable behaviors in the PLDs that can be developed sequentially—and reinforced iteratively for deeper and sustainable learning over time.

The drafts were then discussed among the co-authors (CMR, RET) for "pinpointing", clarifying how different evidence of performance of a given KSA at a given level by anyone developing their stewardship would be exhibited. The "boundaries" between stages relied on Bloom's levels and our own individual experiences with students and colleagues at different stewardship levels. Discussions were both synchronous and asynchronous via online meetings (CMR and RET) and email (CMR, CMG, and RET), to finalize the performance level descriptors.

As each of these three steps were initiated, the interim results were used to triangulate results at other steps in the Mastery Rubric development process. That is, discussions about the PLDs led to the identification of a "missing" KSA (see Results), and also reinforced the choice of the guild structure for the developmental trajectory when concrete descriptions of each KSA at each stage were articulated. Refinements of PLDs for one KSA led to revisions and refinements in other KSA PLDs, to ensure that we pin-pointed performance in terms of stage and KSA, and also that the PLDs were not redundant.

*2.4. Validity Evidence*

Once the Mastery Rubric for Stewardship was created (see Results), case studies were used both to study its validity—as a function of its relevance for professional preparation—and to assess the evidence for our claim that stewardship can be expanded beyond doctoral education to professional preparation: If the MR-S can be used to support training of professionals as well as scholars, there should be considerable alignment between the KSAs and professional practice guidelines—which are intended to guide both scholars and professional practitioners. Degrees of Freedom Analyses were used in all validation, i.e., we aligned the KSAs as predictive of the MR-S with the guideline principles, tabulating in the marginals the number of instances of simple alignment of these features. We did not carry out statistical analysis on the marginals, utilizing only the qualitative assessment of observed alignment to determine whether there was evidence that the MR-S can be useful in training professionals (to behave in concordance with professional practice guidelines). If there was minimal alignment, then we would conclude that the MR-S would *not* be useful for this training—although it might still be useful for training in just stewardship, if not professional practice.

The conclusion that *scholarly* stewardship is supported by the Mastery Rubric for Stewardship (MR-S) was explored with one of three case studies, representing the discipline of History (Case 1). The second case represents the discipline of Statistics and Data Science; both History and Statistics and Data Science have professional practice guidelines that already embrace the professional, as well as the scholarly, practitioner, but History is predominantly a scholarly discipline while Statistics and Data Science comprise a majority of practitioners *outside* of the academy. The third case explores the alignment of the MR-S with neurosciences, which has practice guidelines that emphasize scholarship (rather than general professional practice), although many neuroscience doctorate holders work in, or will go into, industry or other non-academic jobs. By examining the alignment of the MR-S with these three different disciplines, we explored the relevance for the MR-S generally, to determine if different disciplines each need their own MR-S (i.e., with discipline-specific PLDs, which would be suggested if the alignments across these cases were highly variable) or if the MR-S is sufficiently general, which is expected given that the construct of stewardship was intended to be general (i.e., for all doctoral education); this would be suggested if the alignments of the MR-S KSAs with the diverse practice guidelines in these validating case analyses were similar and high.

## 3. Results

### 3.1. Results: Identification of KSAs

As noted, a steward is someone who "will creatively generate new knowledge, critically conserve valuable and useful ideas, and responsibly transform those understandings through writing, teaching, and application". Based on this definition, with consideration of its utility *outside* of doctoral education (i.e., earlier as well as for those who will not pursue a doctorate) as well as within it, the KSAs for stewardship were determined to be:

1.  Requisite knowledge/situational awareness
2.  Create and/or generate new methods/new knowledge
3.  Critically evaluate extant and emerging ideas
4.  Conserve ideas (or not, if deemed rejectable and non-conservation is justified)
5.  Responsibly write (disciplinary scholarship)
6.  Responsibly teach/mentor/model (formally and informally)
7.  Responsibly apply the knowledge and principles of the discipline
8.  Responsibly communicate (outside of scholarly venues)

Seven of the eight KSAs (all except the first item in the list above) were derived directly from the original definition of stewardship. Following the two-phase approach shown in Figure S1 in the Supplementary Materials, while the PLDs for those seven KSAs were being articulated, it became clear that knowing when to exhibit which aspect of stewardship is, itself, something that needs to be taught and practiced explicitly. In the words of the CID, stewardship encompasses both a set of roles and skills, and a set of principles: "the former ensure competence, and the latter provide the moral compass" [1] (p. 9). Thus, we determined that an additional KSA was needed to describe the responsibility to *recognize when* these behaviors need to be applied or modeled, which can represent a concrete and observable version of a "moral compass" (see also [27], p. xxi). We have called this KSA "requisite knowledge/situational awareness" to capture the attention that would be given to standards of professional practice (if they exist) during education or training, or when orienting new employees in the workplace. The applications of the other KSAs of stewardship are contingent upon this situational knowledge. An individual who learns and grows all the other KSAs, but cannot recognize *when to use* them, is less likely to act in a stewardly manner when it is needed. As an individual becomes increasingly inculcated into the habits of mind and practice of a discipline or profession, their abilities to recognize situations in which stewardship is needed should similarly increase. Formally including this KSA in a curriculum is essential for wider exposure to, and greater likelihood of demonstrating, stewardly behaviors.

One of the defining features of the scholarship is the generation of new knowledge; however, modern scientific scholarship, in and outside of academia, can also emphasize new methods or techniques. Thus, we added "generate new methods", which also includes the development of software; computational, mathematical, and statistical techniques, as well as new methods for cross-disciplinary work. Augmenting the KSA this way both broadens the scope of behaviors to which a stewardly approach is important and specifies professional activities (where techniques and methods are developed for industry and non-academic venues) as meaningful sources of evidence of stewardship.

We also broke one of the features of the steward into two separate KSAs: "the critical conservation of valuable and useful ideas" was separated into "critical evaluation of extant knowledge" (KSA 2) and "conserve ideas and justify rejection in non-conservation of ideas" (KSA 3). In fact, critical evaluation of extant knowledge is essential to both the creation of new knowledge and to the conservation (or non-conservation) of the ideas of the discipline. For the scholar, both of these KSAs invariably require a critical evaluation of extant knowledge. However, for the independent practitioner who is not primarily a scholar, conservation (or non-conservation) of ideas may not require a critical evaluation of the literature supporting the idea or an intention to generate new knowledge. One current (2017)

example is the data scientist who creates a new algorithm and, recognizing that the dominant or prevailing computational paradigm creates bias in the algorithm's output (e.g., [28]), rejects that paradigm and seeks to create a different algorithm that does not have that bias. Including two separate KSAs instead of one makes these features of stewardly practice more widely applicable. The differences in performance of these separated KSAs may be less obvious for scholars than for others, but the fact that different performance level descriptors were generated for each KSA affirms the suggested separation. The distinction was further reinforced by recognizing fundamental differences in the two KSAs. Critical evaluation entails an obligation to contribute to the vetting (e.g., by performing peer review) that is essential to promote the vitality and rigor of the literature or knowledge base. Conservation/non-conservation is distinguished with a dimension of understanding *the process* of vetting, and potential influences on vetting by cultural as well as extra-disciplinary forces.

Similarly, we took the "transformation" feature of the steward ("responsibly transform those understandings through writing, teaching, and application") and divided it into four fundamental KSAs, namely: responsibly write; responsibly teach/mentor/model; responsibly apply disciplinary knowledge; and responsibly communicate. In particular, many faculty members are stewardly specifically because they responsibly teach and apply disciplinary knowledge (but do not primarily write scholarly papers), while many non-faculty stewards (e.g., those in industry) responsibly teach informally (e.g., modeling professional practice and mentoring junior members of a team or work group) and communicate outside of scholarly venues. We distinguished "communication" from both teaching and scholarly writing, because some discipline-specific communication (with collaborators, team members, or the public) is *not* considered scholarly, but communication outside of the discipline must be as stewardly as that within the discipline. All of these are key features of stewardly practice, although all practitioners may not have opportunities to develop or demonstrate all four. By separating the components of this stewardship feature, we sought to make each learnable and improvable throughout the development of both scholars and professionals in any discipline.

Thus, these eight KSAs represent the definition of stewardship with subtle refinements that make stewardship achievable and demonstrable by a wider range of practitioners and professionals across disciplines.

### *3.2. Results: Trajectory definition*

As shown in Table 1, four stages of development were articulated based on the European guild structure.

**Novice**: The novice is just beginning to engage with the discipline. This individual is focused on their own performance of any given KSA, which tends to be at Bloom's levels of cognitive complexity 1–2 (understand and summarize). They do not recognize that, or act as if, failures to act in a stewardly manner have ramifications beyond themselves. The novice stage represents the individual embedded in the acquisition of discipline-specific content. This could be an undergraduate declaring the major or an early-stage graduate student.

**Apprentice:** The apprentice is actively engaged in study of the profession or discipline and developing the full range of Bloom's cognitive abilities. She is developing the capacity to practice independently, but has not yet demonstrated ability qualification to do so. However, compared to the novice, her work and KSA performance exhibits greater reflection, and thus, awareness of her own limitations. Additionally, by learning the "tacit responsibility for the quality and integrity of their own work and that of colleagues", and the "responsibility to the larger public for the standards of practice associated with the profession" that Golde and Walker (2006) [1] assume "all professionals" have, the apprentice is aware that professional and disciplinary independence will require stewardship. This stage, which can be the longest period in one's formation, might describe the advanced undergraduate in a major or the later graduate student who has articulated research problems (or their thesis) with some support. However, the stages are not time-dependent, so any individual at any point in a career could be or become an apprentice steward.

**Journeyman**: The journeyman is an independent scholar or practitioner—a steward of the discipline. Depending on the field, this individual could be a doctoral student, an independent scholar (with or without a PhD in the field), or baccalaureate holder who is prepared for independent work in a profession or practice. The journeyman steward ("steward") is uniformly stewardly in their interactions with others in the disciplinary or professional community. They may seek new opportunities to reinforce less-well developed skills. Performance is reflective, and includes analysis and synthesis of their experience with their knowledge (Bloom's 4–6).

**Master**: The Master steward is recognized by evidence and consensus as one who teaches effectively. The distinction between the journeyman and the Master is the Master's demonstrated ability to teach effectively, comprising evidence of successful diagnosis and remediation of the thinking or work of practitioners at earlier stages. (This characterizes performance at the master level in every Mastery Rubric that includes this level. Many independent practitioners (journeymen) have responsibilities to mentor or instruct, but have not accumulated evidence of qualification and the focus on the diagnosis and remediation of challenges that are encountered earlier in development, that distinguishes the Master.) The Master steward is therefore an expert in the KSAs themselves (journeyman) and also as someone to whom apprenticeship in stewardship can be entrusted. Evidence of successful diagnosis and remediation of earlier-stage performance of the KSAs, rather than a listing of the jobs or funding that one's trainees/students have gone on to obtain, is an essential feature of master level stewards. This can include the development and evaluation of assessments that are specific to the KSAs and the progressive evolution of their performance, plus methods by which these cognitive skills can be elicited by, or developed in, those who are less-expert. These would be most clear from a formal educational context but are also important in the workplace: rather than focusing on student work, Master stewards in the workplace might focus on transparency in promotional processes and supporting the creation of evidence that mentees and junior practitioners need in order to be recognized as "developing professionals" who are making progress towards career goals. Simply *modeling* stewardship is not sufficient to train others to become stewardly or to promote active, critical, engagement in the profession or the discipline; thus, achieving the master level requires evidence of the ability to support the development of these KSAs in those whose stewardly behaviors and attitudes are still in development.

### 3.3. Results: Performance Level Descriptors (PLDs)

In the academic or workplace context, the novice steward deals mostly with facts and pre-defined problems, but is also focused on their own actions and is only starting to learn about professional practice standards, such as the fact that standards exist (and why). Novices lack awareness of many or all of the dimensions of stewardship and professionalism, and also of their own development or place in the continuum. In many fields, this corresponds to performance at Bloom's levels 1–2 (understand and summarize), moving towards developing levels 3 (apply) and 4 (analyze/predict) on the stewardship KSAs. The PLDs for the novice are consistently representing these Bloom's levels for all KSAs. Depending on motivation as well as structural support, an individual might spend 1–2 years in this stage (e.g., first two years of college before declaring a major), or longer if they have not begun to identify with a profession or discipline.

The apprentice steward is someone who is actively learning about practice standards as they begin to align themselves and their emerging professional identity with these standards. Their abilities to engage with stewardship KSAs are improving, so they can engage with less-structured problems and less scaffolding. Apprentices perform the KSAs at Bloom's levels 3–4, and because they are learning more about standards of practice for their domain, they would be developing abilities at Bloom's levels 5–6 (evaluate and synthesize) and also demonstrating a growing awareness of their own limitations and opportunities for growth (i.e., developing metacognitive skills). The PLDs for the apprentice represent more confidence with Bloom's levels 3–4 than the novice would be expected to demonstrate for all KSAs, but for those whose sense of professional identity is actively being shaped, engagement

with less structured challenges—requiring Bloom's levels 5–6—would be sought and practiced. In an academic context, this individual collects evidence of their apprenticeship of stewardly behaviors and habits of mind over the period of time "in training" for a job or role. For example, this might be the time spent "in the major" for undergraduates, a master's program, or the majority of the time in a doctoral program. In the workplace, individuals may train for, and learn to identify with, new jobs. However, once an individual leaves the apprentice steward performance level, they will be adaptable to new contexts and might enter this stage briefly, as needed, for refinements to professional identities they are on the way to forming.

The journeyman is a steward of the discipline. They have concrete awareness of professional practice standards and understand how these apply to themselves, to others, and to interactions within and outside of the profession or discipline. The journeyman steward performs the KSAs with understanding, analysis, and synthesis of their experience with their knowledge, functioning at the highest Bloom's levels as they perform all the KSAs that comprise stewardship in their working life. In an academic context, this individual is recognizable as someone to whom the integrity of the field can be entrusted. In the workplace, this individual is recognized as someone whose integrity, and whose commitment to their field or profession, is visibly and concretely demonstrated. One can be considered a journeyman steward when one has evidence of this level of performance on all of the stewardship KSAs that are relevant for their practice; if new dimensions of practice are added (e.g., adding responsibility for junior team members or supervising duties), development of stewardly attributes of those new professional responsibilities can follow the articulated trajectory.

The Master is confirmed, by evidence, as both independent in their performance of the KSAs themselves and also as someone to whom apprenticeship can be entrusted. Master level performance of all stewardship KSAs includes understanding, analysis and synthesis, *and* an understanding of mechanisms by which these cognitive skills can be elicited by those less-expert stewards within their discipline. Evidence of master level performance in stewardship comes from diagnosis (the identification of weaknesses in KSA performance) and remediation (the recommendation of methods to address those weaknesses in the KSAs), including the development and evaluation of assessments that are specific to the KSAs and their progressive evolution. The Master can train others to begin to be, and to be, steward*ly* (novice/apprentice), or to be (journeyman) stewards; they may also train new Master stewards. This individual directly and indirectly supports the development and recognition of stewardly behaviors in those he/she works with whether in academic or workplace contexts. Examples of evidence of master level achievement include the use of individual learning plans that incorporate the development of stewardly behaviors in academic settings, or by encouraging continuing professional development in the workplace. In addition to the explicit "instructional" performance by a Master steward, the Master is also entrusted to support a stewardly context in which the apprentice is trained/prepared for practice. For example, in the academic setting, as the steward creates new knowledge, they understand that the goal of advancing knowledge diverges from the worst characteristics of the "publish or perish" culture, which does not prioritize the discipline and may actually be detrimental to the discipline *and* to public trust in the academic enterprise more generally (see [29]). Outside of the academic setting, where the "publish or perish" attitude may seem absent, there may still be a "rush to results", or emphasis on short-term tasks that can ultimately weaken the discipline or profession. The journeyman steward seeks to ensure that their own work prioritizes the discipline or profession, while the Master steward goes further, seeking to support a culture where the integrity of the discipline is not undermined by pressures such as publication or pushing to complete a project simply for completions' sake or because publication and completion are deliverables.

With these features and characteristics in mind, the specific PLDs were crafted for the KSAs and are presented in the Mastery Rubric for Stewardship in Table 1.

*3.4. Results: Validity Evidence for KSAs of the MR-S*

The Mastery Rubric supports professionals, practitioners, and scholars envisioning themselves as stewards, "committed to the foundation ("heart and essence") of one's field, but also to thoughtful and innovative forward momentum, and development of the future of one's field". To test more empirically whether stewardship can be extended beyond the scholarly disciplinary domain into professional practice (i.e., to test the validity of this extension), we applied the degrees of freedom analysis method to study alignment of the stewardship KSAs with the guidelines for professional conduct—which guide professional and disciplinary practice—in three disparate domains, History, Statistics and Data Science, and Neuroscience. For history, the guidelines are publicly available in narrative form. The key principles for Professional Practice of History were extracted and summarized for the case analysis. For the Professional Practice of Statistics and Data Science, and Neuroscience, the publicly available text for each standard or principle was copied directly from the then-current website (in July 2018). If there is alignment between the Guideline principle and a stewardship KSA, we indicate this with an asterisk (*)—we made no attempt to quantify "how well-aligned" a KSA and Guideline principle is, partly because this will depend on individual training programs' (in schools or the workplace) specific emphasis on either stewardship or their professional standards. Our indications of alignment are, at their most foundational, signals that (or whether) stewardship is consistent with professional practice guidelines.

**Case 1. American Historical Association (AHA) Statement on Standards of Professional Conduct (2018) [30]**

First published in 1987 and most recently revised in 2018, the AHA Standards of Professional conduct recognize that the discipline of history does not belong exclusively to historians. It is an area of "shared human fascination" that is accessible to and produced by people outside of the narrow body of professionals. The Standards thus serve not just as a guide to proper historical inquiry, but also as a way of distinguishing professional historians from others. Professionals are defined by "a self-conscious identification with a community of historians who are collectively engaged in investigating and interpreting the past as a matter of disciplined learned practice". The historian's task begins with discovery. In the words of the AHA, "Scholarship—the discovery, exchange, interpretation, and presentation of information about the past—is basic to the professional practice of history". From here, seven principles describe the "discipline of learned practice":

**(1).** **Critical dialogue:** Historians engage in a complex process of exploring "former lives and worlds in search of answers to the most compelling questions of our own time and place". This process takes place through "reasoned discourse" within "communities governed by mutual respect and constructive criticism".

**(2).** **Trust and respect:** Historians must maintain the trust and respect of readers, both academic and public.

**(3).** **Maintain integrity of the historical record:** Related to the need to maintain trust with the readers is the need to guard the integrity of the historical record. This involves a commitment to not invent, alter, remove, or destroy evidence of any kind, as well as to maintaining the distinction between primary and secondary sources and leaving a clear trail in regards to their use to their use of primary sources and the consistent use of scholarly bibliographies and annotations.

**(4).** **Acknowledging debts:** Trust is also maintained by the proper acknowledgment of one's debts, whether intellectual, financial, or otherwise. This includes avoiding any form of plagiarism, an act of the most serious ethical and professional misconduct. It also includes acknowledging assistance from colleagues, students, or collaborators and or other circumstantial privileges, such as being given special access to material.

**(5).** **Forming points of view:** Among the most basic tasks for historians is the need to form points of view that argue for a "particular, limited perspective on the past". Historians strive to make sense of the past with the recognition that "all knowledge is situated in time and place, that all

interpretations express a point of view, and that no mortal mind can ever aspire to omniscience. Because the record of the past is so fragmentary, absolute historical knowledge is denied us."

**(6).** **Valuing multiple and conflicting perspectives:** At the same time that historians form and defend particular points of view about the past, they also recognize and value differing historical perspectives. This does not mean that all interpretations are equally valid. It means that historians recognize that a final interpretation is impossible—"no single objective or universal account [of the past] could ever put an end to this endless creative dialogue within and between the past and the present".

**(7).** **Recognize personal bias and commit to follow evidence:** Points of view are often shaped by historians' own personal views and biases, thus historians should remain aware of their own biases and commit to following "sound method and analysis wherever they may lead".

Table 2 presents the Degrees of Freedom Analysis matrix to examine the alignment between the Principles of the AHA Guidelines (1–7; columns) and the KSAs of the Steward (Rows). Alignment is indicated with an asterisk.

**Table 2.** Alignment of seven AHA Guideline Principles (columns) with Stewardship KSAs (rows).

| AHA Guideline Principle: MR-S KSAs: | 1 | 2 | 3 | 4 | 5 | 6 | 7 | *ALIGNMENT* |
|---|---|---|---|---|---|---|---|---|
| Requisite Knowledge/ situational awareness | | * | * | | | * | * | 4 |
| Create and/or generate new methods/ new knowledge | * | | | * | * | | | 3 |
| Critically evaluate extant knowledge | * | | | | * | * | | 3 |
| Conserve ideas (or not, if deemed rejectable and non-conservation is justified) | | | * | * | * | * | | 4 |
| Responsibly write | * | * | * | * | * | | * | 6 |
| Responsibly apply disciplinary knowledge | * | | * | | * | | * | 4 |
| Responsibly communicate | * | * | * | * | * | * | * | 7 |
| Responsibly teach/mentor/model | * | * | | * | | * | * | 5 |
| *ALIGNMENT* | 6 | 4 | 5 | 5 | 6 | 5 | 5 | |

Table 2 shows that all of the MR-S KSAs are consistent with at least three of the AHA Practice Standard elements. The KSAs of responsible writing, communicating, and teaching/mentoring/modeling have the strongest alignment (with 5–7 AHA standards); this is not surprising given the fact that History is a profession that is practiced mostly within the academy. According to a study of the 10-year cohort of History PhD recipients in the US published in 2018 [31], roughly a third of PhD graduates between 2004 and 2013 work outside higher education of any sort, while just over half hold tenure track positions in four-year colleges/universities.

When it comes to historians working in non-academic roles, the question of their demonstrating stewardship is more complex. There are a few popular historians, such as David McCullah and Ken Burns, whose work serves an important role in promoting historical literacy within our society. There are also public historians, who work in places such as museums, and institutional historians, who serve on staff at places ranging from the US senate to small organizations and associations. Historians also serve an important role in think tanks or in agencies that maintain or establish archives; including the European Molecular Biology Laboratory (which has outposts, associates, and partner organizations hosting cutting edge science all over the world). These roles, though not in the academy, certainly allow one to bring their historical knowledge, skills, and abilities to bear in their areas of service; therefore, historians both in and outside academia are expected to follow the AHA Practice Standards. The "Alignment" column in Table 2 shows that training the historian in the stewardship KSAs will create opportunities to demonstrate 3–7 of the seven AHA Standards; training individuals to follow the AHA Standards in their practice will create opportunities to demonstrate 5–7 of the eight

KSAs of stewardship. The strong alignment of each AHA Practice Standard with the stewardship KSAs supports the claim that stewardship can be exercised by historical professionals, whether or not they complete a PhD.

**Case 2. The American Statistical Association (ASA) Ethical Guidelines for Statistical Practice (2018) [32]**

The Ethical Guidelines were first endorsed by the ASA Board in 1995 and the latest revision was endorsed by the Board in 2018. These Guidelines include seven general principles and 52 specific elements. The ASA Ethical Guidelines for Statistical Practice describe the professional habits of all practitioners in statistics and data science; practice in government (e.g., Bureau of Labor Statistics; Census), industry (e.g., business; pharmaceutical; biomedical), social science fields (decision making/marketing; industrial/organizational), and scholarship.

**A. Professional Integrity and Accountability**

The ethical statistician uses methodology and data that are relevant and appropriate, without favoritism or prejudice, and in a manner intended to produce valid, interpretable, and reproducible results. The ethical statistician does not knowingly accept work for which he/she is not sufficiently qualified, is honest with the client about any limitation of expertise, and consults other statisticians when necessary or in doubt. It is essential that statisticians treat others with respect.

**B. Integrity of data and methods**

The ethical statistician is candid about any known or suspected limitations, defects, or biases in the data that may impact the integrity or reliability of the statistical analysis. Objective and valid interpretation of the results requires that the underlying analysis recognizes and acknowledges the degree of reliability and integrity of the data.

**C. Responsibilities to Science/Public/Funder/Client**

The ethical statistician supports valid inferences, transparency, and good science in general, keeping the interests of the public, funder, client, or customer in mind (as well as professional colleagues, patients, the public, and the scientific community).

**D. Responsibilities to Research Subjects**

The ethical statistician protects and respects the rights and interests of human and animal subjects at all stages of their involvement in a project. This includes respondents to the census or to surveys, those whose data are contained in administrative records, and subjects of physically or psychologically invasive research.

**E. Responsibilities to Research Team Colleagues**

Science and statistical practice are often conducted in teams made up of professionals with different professional standards. The statistician must know how to work ethically in this environment.

**F. Responsibilities to Other Statisticians or Statistics Practitioners**

The practice of statistics requires consideration of the entire range of possible explanations for observed phenomena, and distinct observers drawing on their own unique sets of experiences can arrive at different and potentially diverging judgments about the plausibility of different explanations. Even in adversarial settings, discourse tends to be most successful when statisticians treat one another with mutual respect and focus on scientific principles, methodology and the substance of data interpretations.

**G. Responsibilities Regarding Allegations of Misconduct**

The ethical statistician understands the differences between questionable statistical, scientific, or professional practices and practices that constitute misconduct. The ethical statistician avoids all of the above and knows how each should be handled.

An 8th Guideline Principle is specific for *employers*:

**H. Responsibilities of Employers, Including Organizations, Individuals, Attorneys, or Other Clients Employing Statistical Practitioners**

Those employing any person to analyze data are implicitly relying on the profession's reputation for objectivity. However, this creates an obligation on the part of the employer to understand and respect statisticians' obligation of objectivity.

Table 3 presents the Degrees of Freedom Analysis matrix to examine the alignment between the Principles of the ASA Guidelines relating to the practitioner (A–G, columns) and the KSAs of the steward (rows). Alignment is indicated with an asterisk.

**Table 3.** Alignment of ASA Professional Statistics Guidelines (columns) with Stewardship KSAs (rows).

| ASA Guideline: Stewardship KSAs: | A | B | C | D | E | F | G | ALIGNMENT |
|---|---|---|---|---|---|---|---|---|
| Requisite knowledge/situational awareness | * | * | * | * | * | * | * | 7 |
| Create and/or generate new methods/ new knowledge | | | | * | * | * | | 3 |
| Critically evaluate extant knowledge | * | * | * | * | * | * | | 6 |
| Conserve ideas (or not, if deemed rejectable and non-conservation is justified) | * | | * | * | * | * | | 5 |
| Responsibly write | * | * | * | | | | | 3 |
| Responsibly apply disciplinary knowledge | * | * | | * | * | * | * | 6 |
| Responsibly communicate | * | * | * | | * | * | * | 6 |
| Responsibly teach/mentor/model | | * | | | * | * | * | 4 |
| ALIGNMENT | 6 | 6 | 5 | 5 | 7 | 7 | 4 | |

Table 3 shows that professional behaviors that are consistent with the ASA Guidelines (columns) can also generate evidence of stewardship of the discipline or profession (rows). The row marginals show that between three and seven of these seven core ASA Ethical Guideline principles are aligned with each stewardship dimension. Thus, although stewardship was originally conceptualized for PhD level practitioners whose primary objective is scholarship, it can also generally support practitioners who are not "scholars first and foremost", which describes many, if not most, practicing statisticians and data scientists. Statisticians and practitioners in quantitative sciences (to whom the ASA Ethical Guidelines pertain, see [32–34]) practice in every discipline, and not solely as scholars; thus, finding alignment between stewardship dimensions and ASA professional practice guideline elements strongly supports the claim that stewardship can be relevant for those who are not principally scholars. Master's level preparation is also sufficient for "professional preparation" as a statistician, and with new undergraduate degree programs in data science being developed across the U.S., it is possible that baccalaureate preparation may suffice for professional preparation in some cases. With the role of data increasing in priority in so many science and technology fields, it is essential to engage all those preparing for statistical practice in active consideration of how they may be as stewardly as possible.

Not every statistician or data scientist who is (or wishes to be) stewardly will have opportunities to teach or create new knowledge; therefore, these "typical" stewardly behaviors cannot be required of every steward of statistics and data science. However, every ASA Ethical Guideline Principle *is* relevant for every practitioner [32–34]—and practicing these principles is clearly relevant for stewardship of the profession of statistics and the practice of data analysis/data science. Importantly, while some students in statistics and data science programs must complete general training in the responsible conduct of research at their universities, that general training, especially if it focuses on principles and practices of research subject protections, can seem unrelated to data intensive applications (see [33,34]). The "Alignment" column in Table 3 shows that training the statistician/data scientist in the stewardship KSAs will create opportunities to demonstrate 3–7 of the seven ASA Guideline Principles relating to the practitioner (A–G); training individuals to follow the ASA Guidelines in their practice will create opportunities to demonstrate 5–7 of the eight KSAs of stewardship. The alignment of each ASA

Guideline Principle with the stewardship KSAs supports the claim that stewardship can be exercised by the full range of professionals in statistics and data science (irrespective of degree completion).

**Case 3. Society for Neuroscience Guidelines for Ethical Practice (2010) [35]**

The Society for Neuroscience (SfN) issued its Guidelines for Ethical Practice in 2010. These Guidelines include nine general principles. The Guidelines describe the scope of practitioners to whom guidelines—and the integrity of its scientific mission—pertain. Importantly, many individuals are included in this scope who are not actually producing scholarship themselves, but are supporting its dissemination (e.g., as editors or reviewers). While most neuroscience is carried out specifically for scholarly purposes, neuroscientists also work in industry (e.g., pharmaceutical; biomedical) and in some social applications (networks), as well as scholarship.

(1). *The integrity of the scientific mission is a collective responsibility.* SfN members and those who contribute to SfN activities and publications are expected to conduct science in a responsible and ethical manner. The institutions at which scientific work is carried out are responsible for ensuring ethical standards are followed. SfN has a special responsibility regarding those scientific activities for which it is directly responsible, including publication of *The Journal of Neuroscience, eNeuro,* and presentations at the annual meeting. Investigators are responsible for the accuracy of information reported in published articles and abstracts, for insuring that authorship is appropriate, for avoiding plagiarism and duplicate publication, and for insuring the ethical treatment of animals and human subjects. Journal editors and reviewers are responsible for providing a fair, objective, and timely process for reviewing submitted manuscripts.

(2). *Data must be original and accurate.* It is essential that researchers and others be able to trust the validity of published data. That trust permits researchers to build on prior observations and thus facilitates the progress of science. Replication and extension of published results allows science to move forward and often entails free sharing of research material. While scientific errors and differences of interpretation are natural aspects of the creative process, data that have been fabricated or falsified contaminate the scientific literature, greatly diminishing its value for researchers and others in the community. Moreover, such fraudulent actions undermine society's trust in the scientific enterprise.

(3). Priority of data and ideas must be respected. Scientific publication is an important part of the process by which priority is established for experimental work and research ideas. Plagiarism—the presentation of other investigator's data or ideas as your own—is unacceptable. Duplication of text or data (including figures, tables, or portions thereof) previously published by others or presentation of ideas or experimental findings of others must be accompanied by citation of the previous work.

(4). *Authorship should reflect a significant intellectual contribution.* Each author should have made a significant intellectual contribution to the conception, design, conduct, analysis, and/or interpretation of the scientific work. Each individual meeting this criterion should be offered the opportunity to participate in authoring, drafting, or critically reviewing the manuscript.

(5). *Original data should only be published once.* Reporting the same finding based on the same data in separate publications without explicit acknowledgement of the relationship constitutes duplicate publication and is unacceptable.

(6). *Every author shares responsibility.* All authors share responsibility for the scientific accuracy of an abstract or manuscript, including supplementary material. Hence, in cases of fabrication, falsification, or plagiarism, all authors are potentially culpable.

(7). *Conflict of interest must be declared.* Authors are responsible for declaring any conflict of interest or appearance thereof that is relevant to a manuscript, abstract, or presentation. Everyone involved in peer review should declare any conflict of interest or appearance thereof and avoid any inappropriate conflict of interest.

(8). ***Pre-published material is confidential.*** Reviewers and editors must avoid breach of confidentiality or using confidential information to advance their own or someone else's research or financial interests.

(9). ***Research using animals and human subjects must be conducted ethically.*** Research using laboratory animals or human subjects must be done humanely and in accordance with institutional and governmental regulations.

Table 4 presents the alignment of the Guidelines from the Society of Neuroscience (1–9, Columns) with the KSAs of the steward (rows). Alignment is indicated with an asterisk.

**Table 4.** Alignment of Neuroscience Guidelines (columns) with Elements of Stewardship (rows).

| Neuroscience Guideline: Stewardship KSAs: | 1 | 2 | 3 | 4 | 5 | 6 | 7 | 8 | 9 | ALIGNMENT |
|---|---|---|---|---|---|---|---|---|---|---|
| Requisite knowledge /situational awareness | * | * | | * | | * | * | | | 5 |
| Create and/or generate new methods/ new knowledge | * | * | * | | | * | * | | * | 6 |
| Critically evaluate extant knowledge | * | | * | | | * | | | * | 4 |
| Conserve ideas (or not, if deemed rejectable and non-conservation is justified) | * | | * | | | | | | * | 3 |
| Responsibly write | * | * | * | * | * | * | * | * | | 8 |
| Responsibly apply disciplinary knowledge | | * | | * | * | * | | | * | 5 |
| Responsibly communicate | | | * | * | * | * | * | * | | 6 |
| Responsibly teach/mentor/model | | | | | | * | * | | | 2 |
| ALIGNMENT | 5 | 4 | 5 | 4 | 3 | 7 | 5 | 2 | 4 | |

Table 4 shows considerable overlap between the KSAs of stewardship and the guidelines of the Society for Neuroscience. The weakest areas of overlap are "conservation of ideas" and "responsibly teach/mentor/model". It is possible that the reason "responsibly teach" is not more explicit in the SFN Guidelines is because most science research is done in academic settings (including research-intensive and medical school contexts), and teaching is a part of the more general "practice" of academia—and as such, might not be included in the specific neuroscience community practice standards (being more akin to "workplace" standards). The conservation of ideas might have less resonance for Neuroscience than for Statistics and Data Science and History if innovation and discovery are higher priorities for neuroscience; in fact, several of the guidelines themselves relate to treating new/breaking knowledge in a stewardly and responsible manner (which is not considered in the other two sets of guidelines).

The "Alignment" column in Table 4 shows that training the neuroscientist in the stewardship KSAs will create opportunities to demonstrate 2–8 of the nine Neuroscience Guideline Principles; training individuals to follow the Neuroscience Guideline Principles in their practice will create opportunities to demonstrate 4–7 of the eight KSAs of stewardship. The alignment of each Neuroscience Guideline Principle with the stewardship KSAs supports the claim that stewardship can be exercised across the professional contexts for the neuroscientist.

## 4. Discussion

As noted in the Introduction, this paper has built upon the work of the Carnegie Initiative on the Doctorate to describe a general curricular framework applicable across disciplines and throughout higher education to promote the explicit formation of stewardship. This Mastery Rubric for Stewardship (MR-S) describes how scholars, professionals, and practitioners, whether inside and outside the academy, can develop and document the characteristics (KSAs) of a steward of the discipline. The three case analyses show considerable—but varying—alignment of the KSAs for stewardship and the professional/ethical practice guidelines and standards for these three disciplines/professions.

The different features of stewardship identified as KSAs can thus be seen to be achievable by *all* practitioners in these diverse fields, and are not limited to those who have completed doctoral training, thus accomplishing the goal of expanding the construct of stewardship beyond the scholar first and foremost. Stewardship can therefore be introduced earlier and more widely—to a far wider audience—than was originally envisioned, and the MR-S describes how performance at each of four stages can be concretely observed—and elicited—in the developing steward. The alignment of each of the KSAs with at least some of the professional practice guideline principles across diverse disciplines suggests that stewardship can be learned and exercised by the full range of professionals across fields as diverse as those we analyzed here. Moreover, focus on the stewardship KSAs during professional/pre-professional training is aligned with these professional practice guidelines.

The purpose of the paper was to broaden the definition of "stewardship" so that it could apply to professionals and practitioners as well as scholars, and to describe a developmental trajectory that can be initiated earlier than doctoral education and at any point in a career so that the broadening could be concretely described. The MR-S was iteratively developed by articulating the KSAs and drafting/revising the PLDs once the stages were identified. During this process, as can be seen in Table 1 (the MR-S):

a.   The cognitive task analysis extracted additional dimensions (KSAs) beyond the original definition of stewardship (generation, conservation, transformation), and eight different features (KSAs) of stewardship that are each learnable and improvable were articulated so that each maintained the essence of the construct. Thus, the construct itself is conserved, and each dimension is rendered learnable, improvable, and observable.

b.   PLDs were formalized to capture a variety of ways that stewardship can evolve organically across different disciplines and throughout a career. They are also general enough so that any KSA at any stage can be demonstrated within and outside of academia, and across fields. Consistent with the original intent of the construct, these PLDs enable any professional or practitioner to demonstrate their stewardship.

c.   PLDs recognize that the journeyman steward may have opportunities to teach, mentor, and model stewardly behaviors, but that specific successes in these activities are concretely demonstrated by the Master. Outside of an academic setting, journeyman stewards can demonstrate their achievement of the master level with the same kinds of evidence as instructors, even if they are derived from non-academic activities, from working with their mentees/junior collaborators. This explicitly supports the development of stewardship in professional settings (outside of academia).

d.   There is no KSA for "ethical practice" because the entire stewardship model implicitly reflects a virtue ethics approach to professional conduct and identity. The focus in the MR-S is on taking, and demonstrating, responsibility in the dimensions of stewardly practice, enabling ethical practice even if there are no/no specific ethical practice guidelines available.

### 4.1. Development of the Stewardly KSAs

The CID's construct of stewardship focused exclusively on doctoral education, and we have agreed that stewardship is most fully and explicitly formed at this level. This is particularly true for the traditional academic disciplines, as well as those where highly specialized knowledge is required to create and critique new ideas. Nevertheless, the MR-S shows that, and how the characteristics and commitments of professional and disciplinary stewardship can be fostered earlier than the doctoral level. Since many professionals engage in disciplinary practice without pursuing the terminal degree, instilling stewardly attitudes at the undergraduate and masters' level would benefit the discipline overall. For example, it has been argued [36] that undergraduate statistics and data science majors can be oriented to the importance of stewardship generally, even if their engagement with the discipline or profession entails neither producing nor critically consuming scientific argumentation. Fostering similar attitudes in other disciplines would benefit not only the disciplines individually but also society more broadly. Importantly, although the first KSA, "requisite knowledge", encourages the

steward-in-training to explore "professional practice standards", these may not exist, may be out of date, or may simply be too specific to the particular profession (e.g., historian, statistician, or neuroscientist) for a modern professional (who may do historical analysis one day, statistical analysis another day). This KSA does not require the professional to rely, or rely solely, on one set of standards but rather, to be aware of those that exist and their relevance in practice.

Since the performance level descriptors of the MR-S are based on Bloom's taxonomy, the Rubric *can* be used with students and professionals across levels. Indeed, since the highest levels of Bloom's taxonomy include "evaluation" and "synthesis", two key characteristics of stewards, all levels of higher education that develop these higher-order cognitive abilities would be able to promote stewardship without dramatically altering coursework or assessment. However, without attention to the growth and development of the characteristic KSAs that define the steward, it is unlikely that higher education can prepare all practitioners to be those to whom the integrity of their respective fields can be confidently entrusted. The MR-S can be used in continuing education and other standard training initiatives across workplaces, if that responsibility is taken up by employers; it can also be used by the self-directed learner to demonstrate their intention and commitment to be stewardly.

In addition to its support of the plausibility of integrating stewardship earlier in education than doctoral level training, the MR-S also represents a curricular structure that is general and flexible enough to be applied across disciplines and institutions. The results of the degrees of freedom analyses showed considerable alignment between the KSAs of stewardship and the professional practice standards from three diverse fields. This alignment suggests that the framework is applicable across many areas of study. In particular, curricula that incorporate disciplinary guidelines can use stewardship to underpin efforts to ensure a developing engagement with the discipline and document the achievement of these pre-professional behaviors. If stewardship were adopted as part of "general training" or general education across a college or department, all students would gain experience in their roles as future stewards of their profession or field. Implementation research can be easily envisioned, with "percent documenting journeyman-level performance" using a portfolio approach, as an outcome to be compared across cohorts from different disciplines or training programs.

As such, the MR-S has the potential to enhance the goals of professional and academic societies and to promote the explicit integration of stewardship at any level or career stage. Many other disciplines (but not all) have published codes of professional conduct. These codes represent the profession and describe how professionals in the discipline can and should engage in their craft (see, e.g., [36–38]). While these disciplinary guidelines and codes all serve to promote the integrity of their respective fields, they are essentially focused on the individual practitioner. The concept of a disciplinary steward may be implied in these codes, but it is never made explicit. A formal consideration of how to promote stewardship in a discipline through the discipline's own model of professional behavior would therefore support the explicit incorporation of both stewardship and its development into the process of training the next generations of scholars *and* professionals. The incorporation of the developmental path to journeyman or Master level stewardship can also facilitate and promote engagement with disciplinary guidelines from early in training. Professional associations have an interest in attracting new members, but the professions themselves have a vital interest in inculcating new members of the profession with the habits of mind that are necessary to promote successful engagement in the domain and between domains (in multidisciplinary work) or between their domain and the public and other stakeholders. Any individual, regardless of where they are in their career, can begin to curate evidence that they possess the KSAs at each stage of their development (e.g., [15]). While we hope the stewardship construct will be taught and practiced more universally in doctoral education, the MR-S can also serve disciplines and professions as they train practitioners prior to and within doctoral programs.

*4.2. Documenting Teaching and Learning*

The MR-S's potential to serve as a general curricular framework lies both in its usefulness in developing scholars, professionals, and practitioners, and its utility for documenting their development.

In this way, the MR-S, as with other Mastery Rubrics, is consistent with recent calls to better document and more effectively communicate learning outcomes in higher education. In 2016, the National Institute for Learning Outcomes Assessment (NILOA) published guidelines for learning outcomes targeting undergraduate education [39]. In 2017, the Council of Graduate Schools (CGS) published the findings of a study of the use of learning outcomes in doctoral education [40]. The 2017 report noted widespread use of learning outcomes among CGS members, as well as growing interest of accrediting bodies in documenting and assessing these outcomes [40] (p. 2). CGS defined a doctoral degree framework as "as set of reference points that defines general skills and competencies of all doctoral recipients" [40] (p. 9). The MR-S (like all Mastery Rubrics) begins with KSAs, and not skills or competencies, but also provides guidance about how "general skills and competencies", if they exist, can be developed in learners [41]. Thus, the MR-S, like other Mastery Rubrics (see [41]) fits and possibly expands on this CGS description. New research to demonstrate how the MR-S meets, and supports meeting, these guidelines, and particularly how implementation of the MR-S in training/education across disciplines can meet the new recommendations, are possible directions for studies of the MR-S and its utility.

Since a key attribute of any Mastery Rubric is flexibility with respect to the source of evidence used to support a claim of achieving a given KSA, the MR-S is well suited for the formal documentation and evaluation of professional development in a variety of settings. Just as a teaching portfolio holds evidence curated to document an individual's attitudes and growth relating to the specific features of education, a "stewardship portfolio" could be curated to document one's attitudes and growth related to their discipline. Those who document their performance of a given (target) KSA, with evidence that is appropriate to their field, would be recognized as achieving that stage. Those who compile this evidence at the journeyman level are designated a "Steward" of their discipline. Those who further meet the Master-level descriptors have demonstrated their expertise and experience in actively and successfully forming earlier-stage stewards. A portfolio approach to the documentation of achievements in stewardship would require the identification and qualification of cohorts of Master level stewards, and a portfolio-based training and assessment program based on the MR-S would support the development and recognition of these cohorts. The potential for faculty development and the strengthening of teaching portfolios could also support faculty buy-in to efforts to integrate the MR-S throughout a curriculum [42]. The stewardship KSAs are opportunities, or suggestions for how to find or create opportunities, to demonstrate accomplishments as well as learning plans. Independent (preferably Master-level) evaluators should agree that evidence supports claims about performance stage achievements (of any KSA) in stewardship. The flexibility in the MR-S arises from the types or sources of evidence that an individual can use to justify claims of achievement, and *portfolios* support this flexibility.

### 4.3. Limitations of this Project

Weaknesses must be acknowledged in the development of the MR-S and particularly in our validation analyses. The primary consideration is that this project was inherently multidisciplinary but does not capture a consensus from any of the three case study disciplines on the alignment between the disciplinary standards and the KSAs of stewardship. This is particularly true for history, since the guidelines themselves were interpreted (by CMR) before they were aligned with the KSAs. We hope and encourage further exploration of the applicability of the MR-S KSAs and their developmental trajectories within these and other disciplines, so that consensus on alignment, as well as dissemination and endorsement of an emphasis on cultivating stewardship, can be developed. In addition, while the MR-S meets or addresses the NILOA and some of the CGS criteria, it is silent on two specific areas of concern regarding the doctoral degree: assessing the need for and quality of doctoral degrees; and improving public understanding of the value of the doctorate. However, by design, the MR-S would specifically increase transparency and understanding for students of the implicit expectations of degree programs, albeit with respect solely to standards of practice. The MR-S is intended to be used to create a "contract" between the instructor and the learner—making program requirements more student-centered by engaging the student in the identification of opportunities to learn, grow,

or demonstrate each KSA at the desired level. It can also support better alignment of training with career paths, particularly with respect to standards of practice along any selected path.

A second consideration is that we did not observe perfect alignment with Guidelines/Standards and the KSAs of stewardship. However, since stewardship was proposed wholly independently of the three sets of practice standards we analyzed, finding any alignment at all tends to support our assertion that stewardship KSAs can support professional practice whether or not the individual will be a "scholar first and foremost", or has/will obtain a PhD. Since not all doctorate holders will be "scholars first and foremost" (many go into industry and government, where scholarship is not a principal priority), at least some divergence from the MR-S KSAs and the professional guidelines is expected. Moreover, not every practitioner or professional can identify with a single set of professional guidelines, and some professions do not have practice standards. The MR-S can help *all* to meet the definition of the steward: "one to whom the integrity and vigor of the discipline can be entrusted". Our alignment results suggest that the stewardship KSAs can support the development of professionalism across a wide range of disciplines, suggesting opportunities for both training and research to test this hypothesis.

The DoFA matrix results do differ by discipline, and each shows some gaps. Some attributes of stewardship are more consistent with some disciplinary guidelines than others. For example, the guidelines for Neuroscience (2010) [35] are highly focused on publication and the creation of new knowledge (i.e., "scholarship"), while the guidelines for Statistics (2018) [32] are mainly focused on data and decision-making. The gaps in the alignment of stewardship with disciplinary guideline principles suggests that disciplinary stewardship is *not redundant with,* and is actually supportive of, the professional ideals for conduct for these disciplines. From this observation, we conclude that stewardship is consistent with the core ideals of these professions. Critically, none of these three disciplines systematically inculcates their professional guidelines into doctoral or pre-doctoral training. Because membership in each disciplinary organization is *optional*, their practice guidelines, while consensus-based and reflective of ethical and professional conduct by all practitioners, cannot be mandated. If educational programs integrate stewardship into their completion requirements, then even superficial engagement with some professional guidelines, and with stewardship, *can* be required by institutions that accomplish this integration.

## 5. Conclusions

The MR-S has the potential to connect the goals of higher education with the formation of professional identity that is intended when discipline-specific guidelines are formulated and published. The Ethical Codes and Guidelines that exist for disciplines are typically complex and require instruction and practice, and becoming a steward of a discipline also requires attention and focus for initiation and development of the defining characteristics. The MR-S maps observable but flexible performance level descriptions across academic and professional development.

Two main conclusions may be drawn from this project. First, the formation of stewards is a worthy goal for *all* higher education—at doctoral, master, and undergraduate levels. Expanding the stewardship model, and facilitating the development of stewardly habits throughout a curriculum, promote achievement of both the expertise (journeyman) and the mentorship (Master) that are necessary for the ongoing vitality and vigor of each discipline as well as in cross-, trans-, and inter-disciplinary work. Second, the MR-S outlines how curricula could be conceptualized so that they promote the KSAs of stewardship and their development.

While not all practitioners have opportunities to do (or job descriptions that include) all of the elements of stewardship, all practitioners can engage with these elements, and they *should* be engaged with their disciplinary guidelines (if they exist; see [43]) and the most stewardly—or steward-*like*—conduct possible. Practitioners in fields or disciplines that do not have ethical guidelines (e.g., economics) can also use a stewardship model, and its developmental trajectory, to integrate attention to professionalism (or ethics) throughout a curriculum. Inculcating the attitudes and commitments of stewardship in any given major, certificate, or training program has the potential to

encourage the translation of these commitments to any discipline or profession in which graduates may engage.

**Supplementary Materials:** The following are available online at http://www.mdpi.com/2227-7102/9/4/292/s1, Table S1: Cognitive Task Analysis Methodology used for KSA identification. Figure S1: A two-phase application of cognitive task analysis to identify KSAs in the Mastery Rubric for Stewardship.

**Author Contributions:** All authors contributed to construction of the MR-S and to writing the paper. R.E.T.: Conceptualization; Data curation; Formal analysis (SFN, ASA); Investigation; Methodology; Project administration; Validation; Visualization; Writing—original draft; Writing—review & editing. C.M.R.: Data curation; Formal analysis (AHA); Writing—original draft; Writing—review & editing. C.M.G.: Writing—original draft; Writing—review & editing. All co-authors take responsibility for the manuscript contents, in accordance with authorship guidelines published by the International Committee of Medical Journal Editors.

**Funding:** This research received no external funding.

**Conflicts of Interest:** The authors declare that there is no conflict of interest.

## Appendix A  Bloom's Taxonomy

Bloom's taxonomy of cognitive complexity [23] specifies a hierarchy with six levels of cognitive skills or functioning:

(1). Remember/Reiterate—performance based on recognition of a seen example.

(2). Understand/Summarize—performance summarizes information already known/given.

(3). Apply/Illustrate—performance extrapolates from seen examples to new examples by applying rules.

(4). Analyze/Predict—performance requires analysis and prediction, using rules.

(5). Create/ Synthesize—performance yields something innovative and novel, creating, describing and justifying something new from existing things/ideas.

(6). Evaluate/Compare/Judge—performance involves the application of guidelines, not rules, and can involve subtle differences arising from comparison or evaluation of abstract, theoretical or otherwise not-rule-based decisions, ideas or materials.

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
