# Peer review of "The Preparation of Stewards with the Mastery Rubric for Stewardship: Re-Envisioning the Formation of Scholars and Practitioners"

_education, doi:10.3390/educsci9040292_

Round 1

Reviewer 1 Report

The issue is interesting, although it is not clear what is the purpose of the manuscript.

The introduction has several references and deals with doctoral training, job opportunities and other related questions. In the Method section a rubric and a content validation is presented (although it is unclear). Subsequently, it establishes a classification of different university figures and characterizes them according to Bloom's taxonomy. But what is the main objective of the paper? What are the research prospectives? What are the research implications?

In the case of results section, it is difficult to discern what a result is from what is discussion. I think the paper should be restructured to make it easier to read.

Author Response

Responses to reviewers

Education Sciences

Manuscript ID: education-565902

The preparation of stewards with the Mastery Rubric for Stewardship: Re-envisioning the formation of scholars and practitioners

Reviewer 1:

Yes

Can be improved

Must be improved

Not applicable

Does the introduction provide sufficient background and include all relevant references?

( )

(x)

( )

( )

Is the research design appropriate?

( )

( )

(x)

( )

Are the methods adequately described?

( )

( )

(x)

( )

Are the results clearly presented?

( )

( )

(x)

( )

Are the conclusions supported by the results?

( )

( )

(x)

( )

Comments and Suggestions for Authors

The issue is interesting, although it is not clear what is the purpose of the manuscript.

We clarified the purpose by identifying it more clearly in the abstract and introduction. We also sought to clarify the relevance of the work to the special issue on professionalism and professional identity development.

The introduction has several references and deals with doctoral training, job opportunities and other related questions. In the Method section a rubric and a content validation is presented (although it is unclear). Subsequently, it establishes a classification of different university figures and characterizes them according to Bloom's taxonomy. But what is the main objective of the paper? What are the research prospectives? What are the research implications?

We clarified the objectives in the introduction, and added considerations and implications for research in the discussion.

In the case of results section, it is difficult to discern what a result is from what is discussion. I think the paper should be restructured to make it easier to read.

We revised the results section so that methods and results are now more clearly delineated, although decisions that were made – particularly with respect to the PLD drafting – necessitate a narrative presentation. We tried presenting the MR-S before the narrative describing the KSAs, stages, and PLDs were derived *after* the MR-S itself, but the section was not clearer with that organization. However, a section of the results has been shifted into the discussion, hopefully clarifying the results section.

Reviewer 2 Report

This is an interesting enough attempt to rethink the process of the intellectual, academic, and professional formation of scholars and practitioners through the application of the Mastery Rubric model in the context of the concept of the stewardship of an academic/professional discipline.

The authors develop a Mastery Rubric for Stewardship (MR-S), and test it against the professional practice standards of three disciplines: History, Statistics and Data Science, and Neuroscience; their specific focus, and the main thrust of the argument, is that their approach stresses the comparability of the conventional academic route of doctoral training and a more applied, professional-practice-based route to the development of stewardship qualities and competencies.

The argument is presented clearly and logically, and it is broadly persuasive; now that the authors have corrected their presentation of the outcomes of their comparison of their Stewardship KSAs against the professional standard statements of their three test disciplines, the paper does not in my view need much further editing, except perhaps to ensure that section and sub-section headings are presented, in the final version of the text, more clearly and consistently than they are in the working version that has been submitted for review.

Author Response

Reviewer made no suggestions, but manuscript was revised per Reviewer 1.

Reviewer 3 Report

The present study investigated the construct of stewardship so that it can be applied to scholars and non-academic practitioners. The authors describe a developmental trajectory based on Bloom’s Taxonomy supporting cross-curriculum teaching for stewardship of a discipline and a profession. Furthermore, they argue that stewardship is an obtainable by all professions. They created a Mastery Rubric for Stewardship that articulates how stewardly behavior can be cultivated and documented for individuals in any disciplinary curriculum. The authors conclude that their mastery rubric for stewardship can be used for curriculum development or revision to promote stewardship in higher education.

This is a well-written and interesting manuscript that I enjoyed reading. The authors provide a good overview of the literature and the problem that they try to solve. However, I have some (minor) concerns that the authors may address in a revised version of their manuscript. First and foremost, using the framework of Bloom’s taxonometric cognitive abilities for each KSA makes perfectly sense. However, in recent years more and more researchers critized Bloom’s model and provided alternative views. Moreover, many studies on the development of expertise also deviate from Bloom’s views. I was a bit surprised that in the limitations section of the manuscript, nothing was mentioned about these criticisms or alternative models.

I am not an expert in qualitative methods, but while reading the Method section, I was struggling to understand what exactly the authors did. Perhaps it was my lack of knowledge but maybe the level of description was too general in this section to be able to reconstruct the authors’ approach.

Finally, and in contrast to my second point, some parts of the manuscript are a bit too elaborate. For instance, given that Blooms’ taxonomy is so widely known, I do not think that it is necessary to discuss this in such a detail as the authors did.

All in all, a very interesting study that will most likely have a strong appeal to many readers of Education Sciences.

Author Response

This is a well-written and interesting manuscript that I enjoyed reading. The authors provide a good overview of the literature and the problem that they try to solve. However, I have some (minor) concerns that the authors may address in a revised version of their manuscript. First and foremost, using the framework of Bloom’s taxonometric cognitive abilities for each KSA makes perfectly sense. However, in recent years more and more researchers critized Bloom’s model and provided alternative views. Moreover, many studies on the development of expertise also deviate from Bloom’s views. I was a bit surprised that in the limitations section of the manuscript, nothing was mentioned about these criticisms or alternative models.

While the reviewer makes a good point – that there are diverse representations of the development of expertise and that Bloom’s model has been both criticized and revised, all of these models (including Bloom’s) describe a progression. Not all describe that progression in observable terms, which is why we utilized Messick’s criteria so heavily in the performance level descriptors. We did not mention the utilization of Bloom’s taxonomy as a limitation because we consider it a strength, and not a limitation. Anderson & Krathwohl (2001 in their revision of Bloom’s 1956 taxonomy, including review of 19 other frameworks) and Moseley et al. (2005 in their review of 41 different frameworks for thinking) survey multiple alternative approaches, revisions, and arguments – but none of them is deemed better than any other – which is another reason for both utilizing Bloom’s and also not considering it a limitation.

I am not an expert in qualitative methods, but while reading the Method section, I was struggling to understand what exactly the authors did. Perhaps it was my lack of knowledge but maybe the level of description was too general in this section to be able to reconstruct the authors’ approach.

We have added a figure to the supplemental materials to support understanding of what exactly was done to create the Mastery Rubric for Stewardship. We have also augmented the discussion of how the degrees of freedom analyses were done in the methods section.

Finally, and in contrast to my second point, some parts of the manuscript are a bit too elaborate. For instance, given that Blooms’ taxonomy is so widely known, I do not think that it is necessary to discuss this in such a detail as the authors did.

We felt that the narrative dedicated to elaborating on Bloom’s taxonomy was necessary because it is not well known at all outside of the United States. Colleagues and collaborators around the world have never heard of it or used it, and since it plays such a key role in the creation of performance level descriptors, it is essential to readers’ understanding of the role of hierarchy in the developmental trajectory that is described by the stages and PLDs.

References:

LW Anderson & DR Krathwohl (Eds). (2001). A Taxonomy for Learning, Teaching, and Assessing: A Revision of Bloom's Taxonomy of Educational Objectives, Complete Edition. New York: Addison  Wesley Longman.

Moseley D, Baumfield V, Elliott J, Gregson M, Higgins S, Miller J, Newton DP. (2005). Frameworks for Thinking. Cambridge, UK: Cambridge University Press.

Round 2

Reviewer 1 Report

The manuscript has improved considerably since the last review. Authors have made most of the recommendations suggested in previous review phases.
However, some spell checking and orthographic errors are still observed. It would be advisable to review the manuscript again.
Results section have improved, although the use of number bullets to expose them makes reading more difficult.
It would be advisable to remove the number bullets and develop whole paragraph ideas.

Author Response

However, some spell checking and orthographic errors are still observed. It would be advisable to review the manuscript again.

The paper has been revised.

Results section have improved, although the use of number bullets to expose them makes reading more difficult. It would be advisable to remove the number bullets and develop whole paragraph ideas.

We have removed the bullet points in the results.